# 🔪KNIFE: Distilling Reasoning Knowledge From Free-Text Rationales

## Abstract

Language models (LMs) have yielded impressive results on many language reasoning tasks, but their unexpected errors raise doubts about their reasoning abilities. In light of this, there is growing interest in finetuning/prompting LMs with both task instances and their associated free-text rationales (FTRs), which explain the correct reasoning process for predicting the correct task output (*i.e.,* how to be "right for the right reasons"). However, existing finetuning methods fail to improve LM performance, while prompting needs prohibitively large (*i.e.,* >50B) LMs to work well. We propose KNIFE, which shows that reasoning knowledge can be effectively distilled from FTRs into a small (*i.e.,* <1B) LM and improve the LM's performance. First, KNIFE finetunes a teacher LM (given task input and FTR) to predict the task output, transferring reasoning knowledge from the FTRs to the teacher's hidden states. Second, KNIFE finetunes a student LM (given task input only) such that its hidden states are aligned with the teacher's. Thus, the student is endowed with reasoning knowledge but can be used for inference without direct FTR input. On two question-answering benchmarks, KNIFE outperforms various finetuning and prompting baselines in fully-supervised and low-resource settings. Also, we observe that FTR quality is a crucial factor in KNIFE's performance.[1]

## 1 Introduction

Whereas conventional supervised learning only gives feedback on a language model's (LM's) task output correctness, *explanation tuning* aims to teach LMs to be "right for the right reasons" (Ross et al., 2017) by also providing them with explanations of the correct reasoning process behind a given correct output (Narang et al., 2020; Hase & Bansal, 2021; Joshi et al., 2022). In particular, there is growing interest in learning from *free-text rationales* (FTRs) — a.k.a. natural language rationales — which use natural language to verbally explain the correct reasoning process for solving a given task instance (Camburu et al., 2018; Rajani et al., 2019; Narang et al., 2020; Wei et al., 2022b).

Among explanation tuning methods (§A.1), the self-rationalization paradigm has been most successful in improving LMs' performance on reasoning tasks (Hase & Bansal, 2021; Wei et al., 2022b; Lampinen et al., 2022). Here, the LM is prompted or finetuned to jointly generate the task output and FTR (Liu et al., 2018; Narang et al., 2020; Marasović et al., 2022; Brahman et al., 2021; Wei et al., 2022b; Zelikman et al., 2022). Although prompted self-rationalization can improve task performance in certain situations (Wei et al., 2022b; Lampinen et al., 2022; Zelikman et al., 2022), prompting typically requires prohibitively large-scale (*i.e.,* >10B) LMs to work well (Wei et al., 2022a). Meanwhile, small-scale (*i.e.,* <1B) LMs are suitable for finetuning, but finetuned self-rationalization is mostly used in the context of explainability and fails to consistently improve task performance (Hase & Bansal, 2021; Zhang et al., 2023). Since few AI researchers or practitioners have the computational resources to freely experiment with their own large-scale LMs (Zhao et al., 2023), how can we use FTRs to improve the performance of small-scale LMs on reasoning tasks?

LMs learn copious latent information during pretraining, but pretrained LMs may not always know how to properly utilize this information to solve reasoning tasks. We refer to knowing how to sensibly synthesize this latent information to solve tasks as *reasoning knowledge*. We hypothesize that the LM often already possesses the latent information needed to solve reasoning tasks, but just lacks the

---

[1]Code and data have been submitted and will be publicly released after the review process has concluded.

reasoning knowledge to organize this information as a meaningful path between the task input and the correct output. Fortunately, FTRs can provide good examples of such reasoning paths. Because an individual FTR explains the reasoning process for solving a single task instance, it only provides *instance-level* reasoning knowledge. Thus, given a set of task instances that sufficiently characterizes the task, it follows that a set of FTRs for these instances can collectively capture *task-level* reasoning knowledge that generalizes to unseen task instances. With this in mind, our goal is to finetune a small-scale LM by extracting general reasoning knowledge from a set of FTRs, then injecting this knowledge into the LM to guide its reasoning and improve its inference performance (Fig. 1). Since FTRs can be feasibly annotated for thousands of instances, which is the typical size of a downstream NLP training set (Mihaylov et al., 2018; Geva et al., 2021), we can reasonably assume that annotated FTRs are available for all finetuning instances. Since we consider human-annotated FTRs, we also assume that all FTRs convey a correct reasoning process and are inaccessible to the LM during inference, as producing the correct FTR requires already knowing the correct task output.

We propose **KN**owledge **DI**stillation From **F**ree-Text Rational**E**s (✎ **KNIFE**), showing that reasoning knowledge can be effectively distilled from FTRs into a small LM and improve the LM's task performance. Unlike prior explanation tuning methods, KNIFE compresses FTR reasoning knowledge from its natural language form into a set of LM hidden states, making it easier to control how this knowledge is transferred. First, KNIFE finetunes a teacher LM to predict the task output, given both the task input and an FTR input. This finetuning process aggregates reasoning knowledge across all finetuning instances' FTRs, then stores it in the teacher's hidden states. Since the teacher requires FTR inputs, it *cannot* be used for inference, during which FTRs are unavailable. Second, KNIFE finetunes a student LM, given only the task input, to align its hidden states with the teacher's. By treating the teacher's hidden states as soft

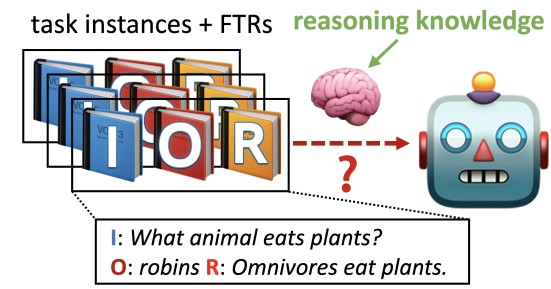

task instances + FTRs    **reasoning knowledge**

**I**: *What animal eats plants?*
**O**: *robins* **R**: *Omnivores eat plants.*

Figure 1: **Reasoning knowledge via FTRs.** Free-text rationales (FTRs) explain the correct reasoning process for solving a given task instance. Meanwhile, a set of FTR-augmented task instances (*i.e.,* [**I**nput, **O**utput, FT**R**] batches) can collectively provide latent reasoning knowledge for solving the task in general. Nonetheless, it remains unclear how to effectively inject this knowledge into LMs to improve their generalization performance.

labels, KNIFE distills reasoning knowledge via such soft labels from the teacher to the student. Then, only the student is used for inference, without FTR input or generation. During inference, instead of the student's output being explicitly conditioned on any specific FTRs, it is implicitly conditioned on the reasoning knowledge that was distilled into the student's parameters.

Traditionally, knowledge distillation (KD) has been studied in the context of model compression, where the teacher's advantage is simply that it is larger than the student (Hinton et al., 2015; Sanh et al., 2019). On the other hand, KNIFE is based on a less common yet still well-studied form of KD called *privileged KD* (Lopez-Paz et al., 2015; Vapnik et al., 2015), where the teacher's advantage is that it has special access to "privileged information" (*e.g.,* additional input features) while the student does not (§A.1). FTR inputs can be considered as privileged information because correct FTRs can be feasibly annotated for thousands of training instances, but are unrealistic to obtain for arbitrary test instances. Therefore, it makes sense to investigate explanation tuning using the privileged KD framework (*i.e.,* teacher gets FTR access, but student does not), as we do with KNIFE.

Our experiments demonstrate KNIFE's ability to improve LM generalization using KD-based explanation tuning. Across two question-answering (QA) benchmarks (OpenBookQA, StrategyQA) and two small-scale LM architectures (T5-Base, T5-Large), we show that KNIFE outperforms various finetuning and prompting baselines on both fully-supervised and low-resource settings, using either human-annotated or model-generated FTRs (§4.4). Also, we validate KNIFE design choices via extensive ablation studies (§4.5). Finally, we analyze KNIFE's failure modes on two additional datasets (ECQA, QuaRTz), identifying FTR quality as critical for KNIFE's performance (§4.6).

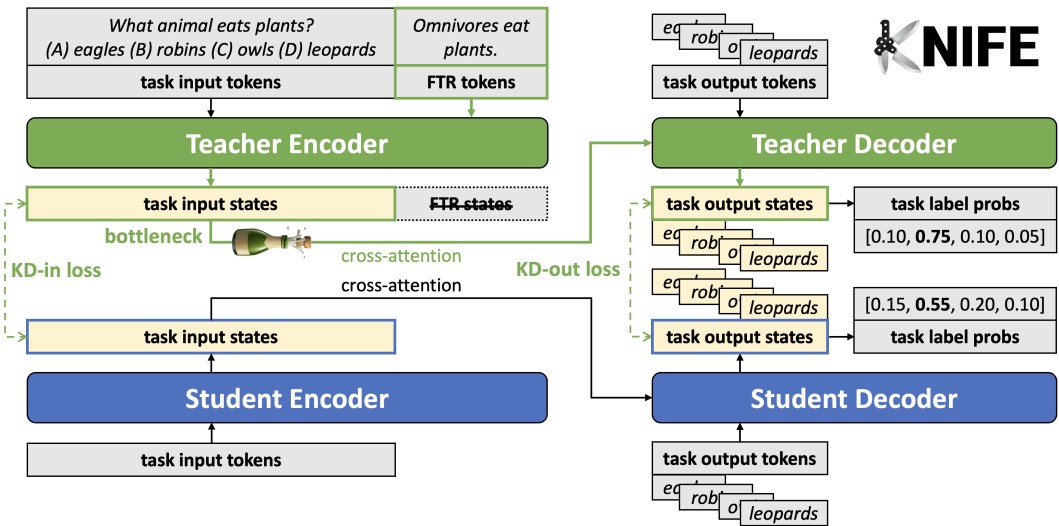

Figure 2: **KNIFE framework.** KNIFE distills reasoning knowledge from an FTR-augmented teacher LM (given task input and FTR) to a student LM (given task input) that is used for inference. The teacher has a *bottleneck*, which masks out all FTR states during cross-attention. As a result, the teacher's decoder must use only its task input (hidden) states to compute its task output (hidden) states, thus routing FTR knowledge to the task input/output states. Finally, FTR knowledge is distilled to the student by training the student so its task input/output states to align with the teacher's.

## 2 BACKGROUND

**Problem Definition** Since reasoning knowledge is latent in the LM's black-box parameters, it is difficult to extract and analyze directly. Thus, for evaluation, our paper follows the established practice of assessing the LM's ability to answer questions designed to require specific reasoning skills (*e.g.,* logical reasoning, commonsense reasoning) (Mihaylov et al., 2018; Geva et al., 2021; Aggarwal et al., 2021; Tafjord et al., 2019). This QA-based classification problem is defined as follows. Given a task input $\mathbf{x}$ (*i.e.,* question) and a set of class labels $Y = \{\mathbf{y}_i\}$ (*i.e.,* answer choices), the model's goal is to predict a score $\rho(\mathbf{x}, \mathbf{y}_i)$ for each $(\mathbf{x}, \mathbf{y}_i)$ pair, so that the predicted label $\hat{\mathbf{y}} = \arg\max_{\mathbf{y}_i \in Y} \rho(\mathbf{x}, \mathbf{y}_i)$ matches the gold (*i.e.,* correct) label $\mathbf{y}^* \in Y$. We consider both multi-choice ($Y$ varies across task instances) and closed-set ($Y$ is fixed for all task instances) text classification.

**Free-Text Rationales** A *free-text rationale* (FTR) $\mathbf{r}$ for a pair $(\mathbf{x}, \mathbf{y}_i)$ is a natural language text that explains the reasoning process for predicting label $\mathbf{y}_i$ (Camburu et al., 2018; Rajani et al., 2019). FTRs could be more intuitive to humans, reference things beyond the task input, and support high flexibility in content, style, and length (Wiegreffe et al., 2021; Chan et al., 2022). Recent works have explored generating FTRs to explain LM behavior (Camburu et al., 2018; Narang et al., 2020; Wei et al., 2022b) and utilizing FTRs to improve LMs (Sun et al., 2022; Wang et al., 2022; Li et al., 2022). We assume each training instance $\mathbf{x}$ is accompanied by an annotated FTR $\mathbf{r}$ for $\mathbf{y}^*$, while FTRs are unavailable for inference instances, as FTR explains the reasoning process and thus indicates the gold label. In this setting, we aim to improve $\mathcal{F}$'s performance by these annotated FTRs.

## 3 KNIFE

We propose KNIFE, an approach to extract reasoning knowledge from FTRs of training instances and inject it into an LM (Fig. 2) using KD. First, KNIFE finetunes a teacher LM to predict the task output, taking as input the concatenation of task input and the FTR. Then, KNIFE finetunes a student LM given only the task input, so that its task encoder/decoder hidden states are aligned with the teacher's. As the teacher and student have different structures of encoder hidden states, the teacher LM has a *bottleneck* design, where the encoder hidden states upon the FTR tokens are masked out in the cross-attention. We are going to elaborate on the method design.

### 3.1 LM DESIGNS

KNIFE consists of a student LM and a teacher LM. They share the same encoder-decoder Transformer architecture and the basic design, with differences in the input and cross-attention mechanism. We are going to first present the basic design, followed by the student and teacher LM designs respectively.

### 3.1.1 Basic Design

We use the encoder-decoder Transformer (Vaswani et al., 2017) architecture for both teacher and student LMs (Raffel et al., 2020; Narang et al., 2020). Building upon §2, we feed the input to the LM's encoder and separately feed each label $\mathbf{y}_i$ to the decoder. For each label $\mathbf{y}_i$, we get the conditional probabilities of all its tokens and then take the average log probability as its score $\rho(\mathbf{x}, \mathbf{y}_i)$. This practice is also adopted by Shwartz et al. (2020) and Wang et al. (2022).

Formally speaking, each label is denoted as $n_{y_i}$-token sequence $\mathbf{y}_i = [y_i^{(1)}, y_i^{(2)}, \dots, y_i^{(n_{y_i})}] \in Y$. By separately teacher-forcing each $\mathbf{y}_i$ to the decoder, we get a conditional probability $P(y_i^{(j)} \mid y_i^{(1)}, \dots, y_i^{(j-1)}, \mathbf{x})$ for each token $y_i^{(j)}$ in $\mathbf{y}_i$. We compute $\rho_i = \rho(\mathbf{x}, \mathbf{y}_i)$ as the score for $\mathbf{y}_i$, by aggregating these token probabilities as:

$$\rho_i = \frac{1}{n_{y_i}} \sum_{j=1}^{n_{y_i}} \log P(y_i^{(j)} \mid y_i^{(1)}, \dots, y_i^{(j-1)}, \mathbf{x}).$$

The predicted probability is calculated by the softmax function, *i.e.*, $P(\mathbf{y}_i \mid \mathbf{x}) = e^{\rho_i} / \sum_{j=1}^{|Y|} e^{\rho_j}$.

### 3.1.2 Student LM Design

The input to the student is always the raw task input, and the student is used for inference. Instead of training the student LM by the cross-entropy loss, we train it to align its encoder/decoder hidden states with those of the teacher LM (§3.2), which is trained before the training of the student. We only consider the hidden states at the top layer in this work. As the token prediction logits are calculated by the LM head taking decoder hidden states as input, we adopt the teacher's LM head for the student.

### 3.1.3 Teacher LM Design

**Main Design**  We aim to first obtain a teacher LM that does reasoning following the FTRs so that its reasoning is guided by reasoning knowledge. To achieve this, we feed the FTRs to the teacher LM. When training the teacher on training instances by task loss $\mathcal{L}_{\text{task}} = -\log P(\mathbf{y}^* \mid \mathbf{x})$, we take as its input the concatenation of the task input and the corresponding FTR. Intuitively, feeding the FTR along with the task input to the teacher LM forces the teacher to reason in a way following the FTR.

Even though the teacher LM's reasoning on training instances is guided by FTRs and thus the reasoning knowledge, we are unable to use it for inference as the FTRs are unavailable for inference instances. However, considering the calculation of its hidden states on a training instance is guided by the corresponding FTR, the hidden states store the FTR knowledge. Following the intuition that a set of FTRs collectively conveys reasoning knowledge, the set of hidden states also collectively conveys reasoning knowledge. Inspired by the works on knowledge distillation (Hinton et al., 2015; Sanh et al., 2019; Jiao et al., 2020, etc), the hidden states of the teacher are also soft labels, on which the student could be trained. By aligning the student's hidden states on training instances with those of the teacher, the knowledge across all FTRs or hidden states is synthesized into the reasoning knowledge conveyed collectively by them, which is finally distilled into the student LM.

**FTR Bottleneck**  In the Transformer architecture, there is a one-to-one correspondence between each token fed to the encoder/decoder and each encoder/decoder hidden state. Each token corresponds to the state upon it. For a specific training instance, the token sequence fed to the student's decoder is identical to that fed to the teacher's decoder. Thus, the student and the teacher share the same structure of the decoder hidden states. We train the student to align each decoder hidden state with the teacher's one upon the same token.

However, the token sequence fed to the student' encoder (the raw task input) is a proper prefix of that fed to the teacher's encoder (the task input concatenated with the FTR). We call the hidden states upon the task input tokens *task input states* and call those upon the FTR tokens *FTR states*. The student and the teacher share the same structure of the task input states but the student doesn't have FTR states as the teacher does. If we just directly ignored the teacher's FTR states in KD, the information stored in the FTR states would be lost. If we fed only raw task input to the teacher in KD, the teacher itself would not work due to the input distribution shift because the FTR is always appended to the input during the training of the teacher.

To address this, the teacher LM has a *bottleneck* design in the cross-attention connecting its decoder to the encoder. Here, the teacher's FTR states are masked out in cross-attention so that the decoder only has access to the task input states. In this way, the training of the teacher funnels the FTR knowledge stored in the FTR token sequence to the task input states by self-attention in the encoder.

As cross-attention is the only path introducing information from the encoder to the decoder, the teacher's FTR states are completely useless now owing to the bottleneck design. Therefore, we can ignore the teacher's FTR states in knowledge distillation as if there had only existed task input states.

## 3.2 KNOWLEDGE DISTILLATION

As both encoder hidden states (task input states) and decoder hidden states store the reasoning knowledge, KD is done by training the student LM so that its encoder and/or decoder hidden states are aligned with the teacher's. We now formally define the learning objectives. Supposing $\mathbf{x}$ is $n_x$-token sequence, the teacher's and student's task input states are denoted as $[\mathbf{e}_{\mathrm{T}x}^{(1)}, \mathbf{e}_{\mathrm{T}x}^{(2)}, \ldots, \mathbf{e}_{\mathrm{T}x}^{(n_x)}]$ and $[\mathbf{e}_{\mathrm{S}x}^{(1)}, \mathbf{e}_{\mathrm{S}x}^{(2)}, \ldots, \mathbf{e}_{\mathrm{S}x}^{(n_x)}]$ respectively. Let dist denote the mean squared error (MSE), a distance function. Let $\mathcal{L}_{\text{KD-In}}$ denote KNIFE's encoder hidden states based KD loss, which pushes the student's task input states ($\mathbf{e}_{\mathrm{S}x}^{(j)}$) to be closer to the teacher's ($\mathbf{e}_{\mathrm{T}x}^{(j)}$):

$$\mathcal{L}_{\text{KD-In}} = \frac{1}{n_x} \sum_{j=1}^{n_x} \text{dist}(\mathbf{e}_{\mathrm{S}x}^{(j)}, \mathbf{e}_{\mathrm{T}x}^{(j)})$$

Similarly, for each label $\mathbf{y}_i$ (fed to the decoder separately), the teacher's and student's decoder hidden states are denoted as $[\mathbf{d}_{\mathrm{T}y_i}^{(1)}, \mathbf{d}_{\mathrm{T}y_i}^{(2)}, \ldots, \mathbf{d}_{\mathrm{T}y_i}^{(n_{y_i})}]$ and $[\mathbf{d}_{\mathrm{S}y_i}^{(1)}, \mathbf{d}_{\mathrm{S}y_i}^{(2)}, \ldots, \mathbf{d}_{\mathrm{S}y_i}^{(n_{y_i})}]$ respectively. Let $\mathcal{L}_{\text{KD-Out}}$ denote KNIFE's decoder hidden states (task output states) based KD loss, which pushes the student's decoder hidden states ($\mathbf{d}_{\mathrm{S}y_i}^{(j)}$) to be closer to the teacher's ($\mathbf{d}_{\mathrm{T}y_i}^{(j)}$):

$$\mathcal{L}_{\text{KD-Out}} = \frac{1}{\sum_{y_i \in Y} n_{y_i}} \sum_{y_i \in Y} \sum_{j=1}^{n_{y_i}} \text{dist}(\mathbf{d}_{\mathrm{S}y_i}^{(j)}, \mathbf{d}_{\mathrm{T}y_i}^{(j)})$$

Finally, let $\mathcal{L}$ denote the total loss defined as $\mathcal{L} = \lambda_{\text{KD-In}} \mathcal{L}_{\text{KD-In}} + \lambda_{\text{KD-Out}} \mathcal{L}_{\text{KD-Out}}$, with loss weights $\lambda_{\text{KD-In}} \in \{0, 1\}$ and $\lambda_{\text{KD-Out}} \in \{0, 1\}$.

## 4 EXPERIMENTS

This section presents experiments.[2] First, in both fully-supervised and low-resource settings, KNIFE outperforms various baselines, using either human-annotated FTRs or model-generated FTRs (§4.4). Second, we validate our KNIFE design choices via extensive ablation studies (§4.5). Third, we analyze KNIFE's failure modes and identify FTR quality as critical to KNIFE performance (§4.6). Finally, for more experiments, please refer to the appendix for extended ablation studies (§A.4), a case study of FTR quality (§A.6), and an FTR rephrasing study (§A.7).

## 4.1 DATASETS

Evaluating LMs' reasoning abilities is still an open problem. Since LMs' internal decision processes are notoriously difficult to interpret, existing benchmarks are generally limited to testing how the LM's outputs vary with different inputs. Thus, our paper follows the established practice of assessing how accurately the LM answers questions that are designed to require specific reasoning skills. We focus on two popular QA datasets: OpenBookQA (OBQA) and StrategyQA.[3] OBQA (Mihaylov et al., 2018) is a four-choice QA dataset that simulates science exams. StrategyQA (Geva et al., 2021) is a boolean QA dataset that requires multi-hop reasoning.

---

[2] We report the mean and standard deviation (std) accuracy over three random seeds for all results, using the format mean ± std. For each table, we use horizontal lines to partition the table into various sub-tables. Each sub-table contains results for methods that have comparable settings (*e.g.,* architecture). That is, Result1 should only be compared to Result2 if there is no horizontal line separating them in the table. In each sub-table, we highlight the best-performing method in red and the second-best performing method in blue.

[3] Since StrategyQA does not provide public test set labels, we use the data split from Wang et al. (2022).

## 4.2 KNIFE DETAILS

**KNIFE Variants**    We consider three KNIFE variants, each with a different combination of weights $\lambda_{\text{KD-In}}$ and $\lambda_{\text{KD-Out}}$ (§3.2). **KNIFE (In)** trains the student LM with $\mathcal{L}_{\text{KD-In}} = 1$ and $\mathcal{L}_{\text{KD-Out}} = 0$. **KNIFE (Out)** trains the student LM with $\mathcal{L}_{\text{KD-Out}} = 1$ and $\mathcal{L}_{\text{KD-In}} = 0$. **KNIFE (In+Out)** trains the student LM with $\mathcal{L}_{\text{KD-In}} = \mathcal{L}_{\text{KD-Out}} = 1$. By default, we use KNIFE (In+Out).

**Implementation Details**    Following prior works (Narang et al., 2020; Sun et al., 2022; Wiegreffe et al., 2021), we use T5-Base and T5-Large (Raffel et al., 2020) as the backbone model for KNIFE and all baselines. For KD-based methods, **T5-Base→T5-Base** means teacher and student use T5-Base, **T5-Large→T5-Large** means teacher and student use T5-Large, and **T5-Large→T5-Base** means T5-Large teacher and T5-Base student.[4]  For non-KD methods used for comparison with the three ones, they use T5-Base, T5-Large, and T5-Base, respectively. In our implementation of KNIFE, if the student and teacher have the backbone model (*i.e.,* both T5-Base or both T5-Large), the student's parameters are initialized as the teacher's. Keeping the model architecture, we also experiment with Flan-T5-Base (Chung et al., 2022), whose initialization parameters are different from those of T5-Base. §A.3 lists the hyperparameters.

## 4.3 BASELINES

We consider a wide range of baselines, spanning various existing paradigms.

**Standard Finetuning** does not involve FTRs or KD. Without any use of FTRs, **FT (I→O)** finetunes a T5 on the task dataset by the cross-entropy loss.

**Finetuned Self-Rationalization** inserts an FTR to the LM's target output during training. **FT (I→OR)** finetunes a T5 to generate the task output followed by the FTR. **FT (I→RO)** finetunes a T5 to generate the FTR followed by the task output. They both take as input the raw task input.[5]

**Input Augmentation** appends an FTR to the LM's input during training. **FT (IR→O)** finetunes a T5 to predict the output taking as input the task input concatenated with the corresponding FTR. This is equivalent to training KNIFE's teacher LM without the bottleneck design. As the inference instances do not have FTRs, the input during inference is just the raw task input. The global patterns that FTR is appended to the training input are absent during inference, which causes the issue of input distribution shift. **FT Dropout (IR→O)** randomly drops out the appended FTR during training, in order to mitigate input distribution shift by also training the model to deal with the raw task input.

**Prompted Self-Rationalization** uses chain-of-thought (CoT) prompting (Wei et al., 2022b). **CoT (I→RO)** prompts GPT-NeoX and GPT-3 (text-davinci-003) to generate the FTR followed by the task output by CoT prompting.[6]

**Pipeline Rationalization** finetunes two LMs as a pipeline. For **FT (I→R→O)**, the first T5 is finetuned to generate the FTR given the task input, while the second T5 is finetuned to generate the task output given the first T5's generated FTR.[7] **FT (I→R, IR'→O)** also utilizes the FTR generated by the first T5, and the difference is that the second T5 is finetuned to generate the task output given both the task input and the generated FTR.

**FT Teacher Init.** finetunes a T5 as FT (I→O) does, with initializing the LM with the KNIFE teacher's parameters. Intuitively, the teacher's parameters store reasoning knowledge, so it is natural to view parameter initialization as a way of transferring such knowledge.

---

[4]For T5-Large→T5-Base, the teacher and student have different representation spaces of hidden states. Thus, in KD we jointly train two linear projection layers to transform the student LM's encoder and decoder hidden states to be in the same representation space as the teacher LM's.

[5]As labels other than the gold label don't have FTRs, FT (I→OR) and FT (I→RO) cannot use the basic design in §3.1. Instead, we train the LM by teacher-forcing the target output to the decoder. During inference, we use greedy decoding to generate the prediction.

[6]All CoT (I→RO) results were obtained from Wang et al. (2022), except GPT-3 on OBQA, which was obtained from Huang et al. (2022).

[7]Since this method is known to not perform well (Wiegreffe et al., 2021; Wang et al., 2022), we only consider FT (I→R→O) in a limited set of settings. In these settings, we present the results reported in Wang et al. (2022). Note that only the mean performance is available.

## 4.4 MAIN RESULTS

Table 1 presents our main results. Here, LMs are finetuned on the entire training set. Methods requiring FTRs use the human-annotated FTRs (called gold FTRs in the following paragraphs) provided by the public datasets. We observe that KNIFE (In+Out) consistently outperforms all baselines, suggesting KNIFE effectively extracts reasoning knowledge from FTRs of training instances and utilizes such knowledge for performance improvement. Besides, KNIFE is the only FTR-requiring method that consistently outperforms FT (I→O), which shows the difficulty of improving small models' task performance by FTRs.

Table 1: **KNIFE main results**

| Architecture | Method | Accuracy (↑) | |
| --- | --- | --- | --- |
| | | OBQA | StrategyQA |
| T5-Base→T5-Base | FT (I→O) | 57.93 (±1.15) | 59.05 (±0.23) |
| | FT (I→OR) | 53.93 (±1.33) | 51.84 (±1.45) |
| | FT (I→RO) | 55.53 (±0.46) | 58.65 (±1.53) |
| | FT (I→R→O) | 56.65 | 57.11 |
| | FT (IR→O) | 53.73 (±2.31) | 49.97 (±2.92) |
| | FT Dropout (IR→O) | 58.27 (±1.33) | 55.85 (±2.09) |
| | FT (I→R, IR'→O) | 58.40 (±0.92) | 57.52 (±4.53) |
| | FT Teacher Init. | 58.33 (±0.90) | 57.25 (±2.22) |
| | KNIFE (In+Out) | 61.53 (±0.76) | 60.45 (±0.31) |
| T5-Large→T5-Large | FT (I→O) | 65.60 (±0.40) | 57.58 (±0.70) |
| | FT (I→OR) | 61.93 (±1.97) | 57.58 (±0.12) |
| | FT (I→RO) | 61.87 (±2.12) | 63.66 (±1.14) |
| | FT (IR→O) | 61.27 (±2.16) | 53.24 (±2.54) |
| | FT Dropout (IR→O) | 65.73 (±1.36) | 59.25 (±4.59) |
| | FT (I→R, IR'→O) | 63.53 (±0.83) | 61.46 (±1.82) |
| | FT Teacher Init. | 65.67 (±2.25) | 61.72 (±2.36) |
| | KNIFE (In+Out) | 68.73 (±1.36) | 63.79 (±0.64) |
| T5-Large→T5-Base | Best T5-Base→T5-Base | 58.40 (±0.92) | 58.65 (±1.53) |
| | KNIFE (In+Out) | 60.93 (±0.12) | 61.12 (±2.03) |
| GPT-NeoX | CoT (I→RO) | 33.80 | 55.31 |
| GPT-3 (text-davinci-003) | CoT (I→RO) | 86.40 | 66.53 |

We observe that FT (I→OR) and FT (I→RO) often bring performance drop compared to FT (I→O), which is consistent with observations from prior works (Hase & Bansal, 2021; Zhang et al., 2023). Our explanation is that task and FTR generation objectives could conflict with each other, meaning that jointly optimizing them may hurt task performance. Besides, the generated FTRs could contain hallucinated information misleading the answer prediction (Zhang et al., 2023), which could also account for FT (I→R, IR'→O) being consistently beaten by KNIFE .

FT (IR→O) is significantly worse than FT (I→O), which is expected due to the input distribution shift issue (§4.3). FT Dropout (IR→O) mitigates the issue to some extent but still fails to bring consistent improvement. The results also support the argument that we cannot directly use KNIFE's teacher LM during inference (§3.1), which is basically FT (IR→O) with bottleneck design.

FT (I→R→O) is also worse than FT (I→O). It is because the generated FTR could be poor, omitting critical information from the task input or introducing irrelevant and even incorrect information, which can hurt task performance (Huang et al., 2022; Magister et al., 2022; Li et al., 2022).

FT Teacher Init. outperforms FT (I→O) in most cases, but its improvement is much smaller than that of KNIFE (In+Out). Thus, parameter initialization is a potential way to transfer reasoning knowledge from the teacher to the student, but it is much less effective than knowledge distillation.

Also, we report CoT (I→RO) results for GPT-NeoX (20B) (Black et al., 2022) and GPT-3 (text-davinci-003, 175B) (Brown et al., 2020). Since GPT-NeoX and GPT-3 are much larger than T5-Base (220M) and T5-Large (770M), it is unfair to expect other methods to perform as well as CoT (I→RO). Even so, we find that KNIFE (In+Out) greatly outperforms GPT-NeoX on all settings, while KNIFE (In+Out) with T5-Large achieves similar performance to GPT-3 on StrategyQA.

In Table 4, we repeat these experiments for GPT-NeoX generated FTRs (Wang et al., 2022) and find that KNIFE (In+Out) still consistently outperforms all baselines. It shows KNIFE's robustness to different sources of FTRs. Interestingly, KNIFE with GPT-NeoX FTRs still considerably outperforms CoT (I→RO) with GPT-NeoX, despite KNIFE using much smaller LMs.

In Table 5, we consider a low-resource setting where available is only 10% of the training data, using T5-Base→T5-Base on OBQA. We find KNIFE beats all baselines, showing that KNIFE effectively extracts and injects reasoning knowledge also in the low-resource setting.

In Table 11, we show the results of FLAN-T5-Base. We find that KNIFE still outperforms all baselines on OBQA, but to a lesser degree than for T5-Base and T5-Large. Meanwhile, for FLAN-T5-Base, KNIFE does not yield improvements on StrategyQA. This is expected for two reasons. First, FLAN-T5 is already a very strong model due to its extensive instruction-tuning across 1836 tasks. Second, FLAN-T5 is already instruction-tuned on StrategyQA CoT data in particular. Consequently, there is more limited room for improvement in finetuning FLAN-T5 on individual datasets' FTRs.

We conducted the Wilcoxon rank-sum test between KNIFE and the best baseline for each OBQA/StrategyQA setting. Based on this test, KNIFE achieves statistically significant ($p < 0.05$) improvement over the best baseline in all settings, except T5-Large→T5-Large on StrategyQA.

## 4.5 ABLATION STUDIES

To justify design choices of KNIFE and understand why it works, we present ablation studies, analyzing the impacts of KD objective, FTR usage, FTR perturbation, teacher bottleneck, and student task loss.

**KD Objectives** Table 2 compares the performance of KNIFE (In), KNIFE (Out), and KNIFE (In+Out). For both gold (human-annotated) and GPT-NeoX (model-generated) FTRs, we find that KNIFE (In+Out) generally achieves the highest performance with a few exceptions.

Table 2: **KNIFE variants**

| Architecture | Method | Accuracy (↑) | |
|---|---|---|---|
| | | OBQA | StrategyQA |
| T5-Base→T5-Base | KNIFE (In) + Gold | 60.00 (±0.40) | 61.39 (±1.90) |
| | KNIFE (Out) + Gold | 62.27 (±1.01) | 59.05 (±1.22) |
| | KNIFE (In+Out) + Gold | 61.53 (±0.76) | 60.45 (±0.31) |
| | KNIFE (In) + GPT-NeoX | 61.07 (±0.12) | 61.92 (±1.74) |
| | KNIFE (Out) + GPT-NeoX | 61.60 (±0.53) | 60.72 (±0.20) |
| | KNIFE (In+Out) + GPT-NeoX | 61.53 (±0.76) | 61.92 (±1.04) |
| T5-Large→T5-Large | KNIFE (In) + Gold | 66.20 (±0.53) | 62.66 (±3.38) |
| | KNIFE (Out) + Gold | 68.07 (±1.50) | 64.40 (±1.22) |
| | KNIFE (In+Out) + Gold | 68.73 (±1.36) | 63.79 (±0.64) |
| | KNIFE (In) + GPT-NeoX | 67.20 (±0.40) | 62.32 (±1.84) |
| | KNIFE (Out) + GPT-NeoX | 68.53 (±1.89) | 62.26 (±0.64) |
| | KNIFE (In+Out) + GPT-NeoX | 68.73 (±1.55) | 63.99 (±0.81) |
| T5-Large→T5-Base | KNIFE (In) + Gold | 31.13 (±2.87) | 53.77 (±0.46) |
| | KNIFE (Out) + Gold | 55.60 (±2.42) | 61.12 (±0.53) |
| | KNIFE (In+Out) + Gold | 60.93 (±0.12) | 61.12 (±2.03) |
| | KNIFE (In) + GPT-NeoX | 30.07 (±2.97) | 53.91 (±0.69) |
| | KNIFE (Out) + GPT-NeoX | 55.60 (±2.03) | 60.59 (±0.61) |
| | KNIFE (In+Out) + GPT-NeoX | 60.47 (±0.81) | 62.39 (±0.42) |

This suggests useful FTR knowledge can be distilled via both encode and decoder hidden states, so it is recommended to use both by default. Furthermore, KNIFE (In) and KNIFE (Out) are also able to outperform all baselines in almost all cases.

KNIFE (In) performs much worse for T5-Large→T5-Base than the two others and many baselines. Since KNIFE (In) only performs KD via the encoder hidden states, the student's decoder is not trained. Here, the student cannot be initialized with the teacher's parameters as their backbone models are different, leaving the student's decoder with T5-Base's pretrained parameters. Thus, in such cases, KNIFE (In) is problematic and KD is necessary via the decoder hidden states.

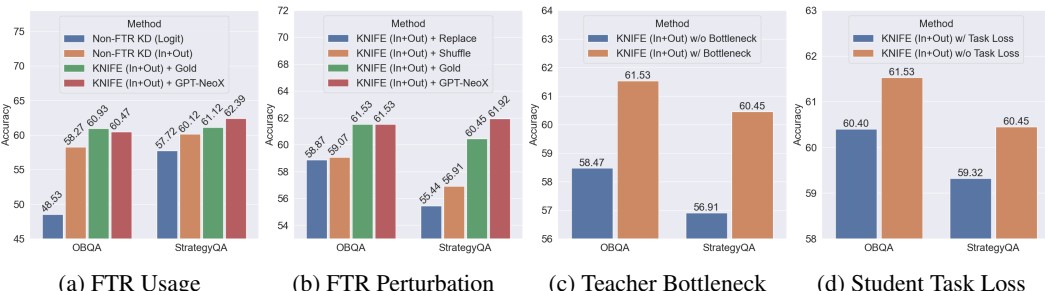

(a) FTR Usage     (b) FTR Perturbation     (c) Teacher Bottleneck     (d) Student Task Loss

Figure 3: **KNIFE ablation studies**

**FTR Usage** KD has been widely used to transfer knowledge from a larger teacher to a smaller student (Hinton et al., 2015; Sanh et al., 2019, etc), so the larger teacher's capacity could be one source of performance gain for T5-Large→T5-Base. To verify FTR usage's importance for the performance gain, we compare KNIFE to non-FTR KD methods, where the teacher is FT (I→O). **Non-FTR KD (Logit)** trains the student to align its logit distribution with the teacher's. **Non-FTR KD (In+Out)** trains the student to align its encoder and decoder hidden states with the teacher's. In Fig. 3a, both KNIFE (In+Out) + Gold and KNIFE (In+Out) + GPT-NeoX outperform the non-FTR KD baselines, showing that KNIFE actually benefits from FTRs and thus the reasoning knowledge. Plus, Non-FTR KD (Logit) performs much worse than Non-FTR KD (In+Out), which validates KNIFE's use of representation-based KD. We also experiment with the self-distillation baseline, where both the student and the teacher are T5-Base and the FTR is not used, *i.e.,* Non-FTR KD between two T5-Base. The results are in Table 13. We find that KNIFE significantly outperforms the Non-FTR KD, which revalidates KNIFE's performance gain is from the FTRs.

**FTR Perturbation**    To prove the performance gain of KNIFE is not from any unexpected noise, *e.g.,* more computation of the teacher LM due to longer input, we perturb the FTRs and observe how it influences the performance of KNIFE. **Replace** replaces each FTR token with a random token from the token vocabulary. **Shuffle** shuffles FTRs across all training instances in the dataset. In Fig. 3b, KNIFE (In+Out)'s performance with Replace and Shuffle is much lower than with Gold and GPT-NeoX, which shows the performance gain actually comes from the FTR knowledge.

**Teacher Bottleneck**    We empirically validate the bottleneck's necessity for KD via encoder hidden states by training a teacher without the bottleneck. In Fig. 3c and Table 8, we show that removing bottleneck brings significant performance drop to KNIFE (In+Out) and KNIFE (In), but it is not as critical for KNIFE (Out). It is expected because the bottleneck of KNIFE's teacher is designed to address the issue that the student and teacher have different structures of encoder hidden states, and KD via only decoder hidden states doesn't meet the issue.

**Student Task Loss**    By default, KNIFE trains the student with only KD losses. We justify it by comparing it to KNIFE variants where the task loss is also involved in the total loss. Specifically, KNIFE with task loss trains the student by the loss $\mathcal{L} = \lambda_{\text{KD-In}}\mathcal{L}_{\text{KD-In}} + \lambda_{\text{KD-Out}}\mathcal{L}_{\text{KD-Out}} + \mathcal{L}_{\text{task}}$. In Fig. 3d and Table 9, we see omitting the task loss consistently yields higher performance. Intuitively, KD loss is guided by the supervision signal of reasoning knowledge, while task loss is guided by the supervision signal of task labels, which could be sub-optimal for learning reasoning knowledge. Thus, the KD and task losses could conflict during optimization, which hurts the performance.

**Teacher Performance**    We evaluate the teacher's test performance both with and without FTR input in Table 12. As expected, the teacher's performance with FTR input is unrealistically high because the FTR is essentially leaking the answer to the teacher. Meanwhile, the teacher's performance without FTR input is much lower due to input distribution shift, as the teacher was trained to take both task inputs and FTRs as input. Actually, its performance is quite similar to that of FT (IR→O) as the teacher model is basically the FT (IR→O) model with the bottleneck design. It also shows that we cannot directly use the teacher for inference, where we assume the FTRs are not available.

## 4.6 Failure Analysis

Although KNIFE performs well on OBQA and StrategyQA, it yields negative results on other QA datasets like ECQA (Aggarwal et al., 2021) and QuaRTz (Tafjord et al., 2019). Using T5-Base→T5-Base, we compare the performance of KNIFE and the main baselines considered in the main results. In Table 3, we see that KNIFE generally outperforms all FTR-based baselines, sometimes by a very large margin. Still, none of the FTR-based methods (including all KNIFE variants) are able to significantly outperform FT (I→O).

Since KNIFE distills FTR knowledge to the student LM, the student's performance is expected to depend on the amount and quality of reasoning knowledge stored in the FTRs. Thus, to investigate these negative results, we conduct a case study to qualitatively analyze the gold FTRs in OBQA, StrategyQA, ECQA, and QuaRTz. Overall, we find that FTRs in OBQA and Strate-

Table 3: **Failure analysis**

| Architecture | Method | Accuracy (↑) | |
| --- | --- | --- | --- |
| | | ECQA | QuaRTz |
| | FT (I→O) | 62.02 (±0.48) | 68.20 (±0.52) |
| | FT (I→OR) | 56.09 (±0.47) | 57.19 (±0.58) |
| | FT (I→RO) | 54.60 (±0.66) | 56.76 (±2.74) |
| T5-Base→T5-Base | FT (IR→O) | 41.02 (±1.57) | 66.41 (±0.90) |
| | KNIFE (In) | 55.12 (±2.19) | 68.45 (±0.83) |
| | KNIFE (Out) | 57.26 (±2.68) | 68.45 (±0.52) |
| | KNIFE (In+Out) | 56.12 (±1.91) | 68.41 (±0.99) |

gyQA are much more informative than those in ECQA and QuaRTz. For OBQA and StrategyQA, we find that their gold FTRs tend to have the following properties. First, they describe a logically sufficient reasoning process for getting from the question (input) to the answer (output). Second, they provide general and self-contained knowledge that goes beyond the information given in the question and answer. Meanwhile, FTRs from ECQA and QuaRTz tend to exhibit opposite properties. They usually simply rephrase the question and answer. To illustrate our case study, we give some examples in §A.6. To further illustrate how common is simple rephrasing among bad FTRs, we conduct an experiment to quantitatively show it in §A.7. Consequently, KNIFE's failure on ECQA and QuaRTz is owing to the uninformative nature of their human-annotated FTRs. Given that, we hope future work could annotate datasets with FTRs informative enough, which would collectively convey useful and sufficient reasoning knowledge and thus contribute to performance improvement.

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

## A  APPENDIX

### A.1  RELATED WORK

**Explanation Tuning**   There are three main existing paradigms for FTR-based explanation tuning: *self-rationalization*, *input augmentation*, and *pipeline rationalization* (Fig. 4). In self-rationalization, the LM is finetuned or prompted to generate both the task output and FTR (Liu et al., 2018; Narang et al., 2020; Brahman et al., 2021; Marasović et al., 2022; Li et al., 2022; Wei et al., 2022b; Zelikman et al., 2022; Lampinen et al., 2022; Majumder et al., 2022). However, finetuning struggles to improve LM performance (Hase & Bansal, 2021) and prior works on it focus on explainability or FTR generation capability. Prompting requires large and often prohibitively large LMs to work well. Besides self-rationalization, the other two paradigms struggle to improve task performance due to some intrinsic issues. Methods under the input augmentation paradigm use FTRs as additional inputs (Rajani et al., 2019; Hase et al., 2020; Kumar & Talukdar, 2020; Wiegreffe et al., 2021). They need to resolve the input distribution shift issue, which occurs when incorporating FTRs into inputs during training but not during inference, and thus struggle to improve task performance. Some other works explored the pipeline rationalization paradigm, where a finetuned rationalizing

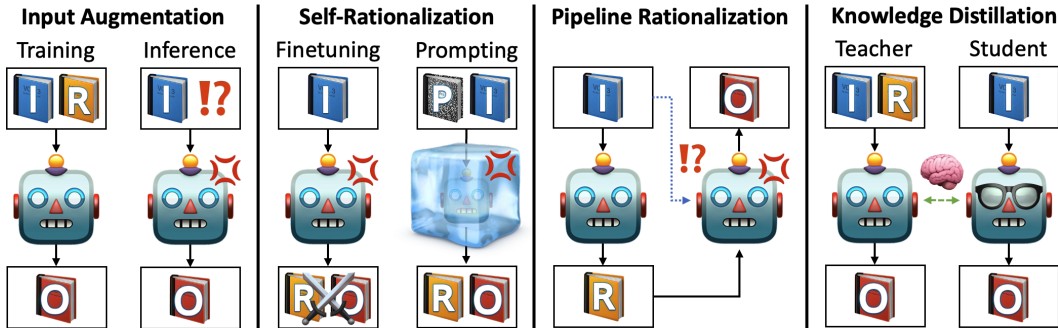

Figure 4: **Explanation tuning paradigms**

LM first generates the FTR and then a finetuned reasoning LM predicts the task output given the generated FTR (Rajani et al., 2019; Kumar & Talukdar, 2020; Hase et al., 2020; Wiegreffe et al., 2021). However, the generated FTR could be poor, omitting critical information from the task input or introducing irrelevant and even incorrect information, which can hurt task performance (Huang et al., 2022; Magister et al., 2022; Li et al., 2022). Plus, the generated FTR forms a non-differentiable path, which complicates end-to-end training.

On the other hand, KNIFE is based on *knowledge distillation*, which addresses the key limitations of existing paradigms. Unlike finetuned self-rationalization, KNIFE does not involve jointly optimizing task and FTR generation losses (which may conflict), since the student is only trained with KNIFE's distillation losses. Unlike prompted self-rationalization, KNIFE does not require large LMs, since the teacher and student are finetuned instead of relying only on their pretrained knowledge. Unlike input augmentation, KNIFE does not need FTRs for inference or cause an input distribution shift, since the student is given only the task input during both training and inference. Unlike pipeline rationalization, KNIFE does not require multiple inference LMs or create a non-differentiable path between them, since only the student is used for inference. Plus, KNIFE generally has lower inference-time costs, since the student does not process additional FTR inputs or generate additional FTR tokens.

**Knowledge Distillation**    Knowledge distillation has been widely used to transfer knowledge from a larger teacher to a smaller student model (Hinton et al., 2015; Sanh et al., 2019; Jiao et al., 2020; Mirzadeh et al., 2020, etc). Instead of aiming for this typical goal, KNIFE distills the FTR knowledge through the hidden states of a teacher model to a student model, which has no direct access to FTRs. Similar to the line of work that incorporates knowledge distillation with privileged information (Lopez-Paz et al., 2015; Vapnik et al., 2015; Fukuda et al., 2017; Wang et al., 2018), where student models benefit from privilege information, the student model in KNIFE essentially gains additional knowledge from FTRs rather than relying on the larger teacher model capacity. Snell et al. (2022) propose to internalize the in-context learning ability such that the performance gains can keep without context tokens. It does not directly distill the knowledge from FTRs and requires dedicated prompt designs. Shridhar et al. (2022); Magister et al. (2022); Ho et al. (2022) propose to distill reasoning abilities from larger language models to smaller models. They require large-scale language models with such abilities, while KNIFE can work well with small models.

## A.2    DATASETS

In this paper, we consider four datasets: OBQA, StrategyQA, ECQA, and QuaRTz. Each dataset tests different kinds of reasoning abilities. OBQA tests the LM's ability to synthesize both science facts (*e.g.,* "metal is a thermal conductor") and common facts (*e.g.,* "steel is made of metal") to answer science exam questions (Mihaylov et al., 2018). StrategyQA tests the LM's ability to decompose a question into multiple steps and use various implicit facts (*e.g.,* "Osiris was the Egyptian god of the underworld") and reasoning skills (*e.g.,* number comparison, set inclusion) to combine the steps into a sensible answer (Geva et al., 2021). ECQA test the LM's ability to recall commonsense knowledge and reason about this knowledge (Aggarwal et al., 2021). QuaRTz tests the LM's ability to understand and answer questions about a wide range of qualitative relationships (*e.g.,* differing comparatives, discrete property values) (Tafjord et al., 2019).

Table 4: **KNIFE main results (GPT-NeoX FTRs)**

| Architecture | Method | Accuracy (↑) | |
|---|---|---|---|
| | | OBQA | StrategyQA |
| T5-Base→T5-Base | FT (I→O) | 57.93 (±1.15) | 59.05 (±0.23) |
| | FT (I→OR) | 50.60 (±1.25) | 53.04 (±1.36) |
| | FT (I→RO) | 49.93 (±3.20) | 56.18 (±2.58) |
| | FT (IR→O) | 48.40 (±1.71) | 49.37 (±2.52) |
| | FT Dropout (IR→O) | 58.53 (±1.14) | 60.39 (±1.17) |
| | FT (I→R, IR'→O) | 53.27 (±2.80) | 55.64 (±2.21) |
| | FT Teacher Init. | 59.80 (±1.64) | 59.25 (±0.42) |
| | KNIFE (In+Out) | 61.53 (±0.76) | 61.92 (±1.04) |
| T5-Large→T5-Large | FT (I→O) | 65.60 (±0.40) | 57.58 (±0.70) |
| | FT (I→OR) | 59.20 (±1.56) | 59.79 (±2.20) |
| | FT (I→RO) | 59.40 (±0.72) | 55.58 (±1.10) |
| | FT (IR→O) | 53.13 (±1.94) | 49.70 (±3.03) |
| | FT Dropout (IR→O) | 66.87 (±0.31) | 59.85 (±1.80) |
| | FT (I→R, IR'→O) | 60.47 (±1.03) | 55.58 (±0.64) |
| | FT Teacher Init. | 66.87 (±1.10) | 59.99 (±1.01) |
| | KNIFE (In+Out) | 68.73 (±1.55) | 63.99 (±0.81) |
| T5-Large→T5-Base | Best T5-Base→T5-Base | 59.80 (±1.64) | 60.39 (±1.17) |
| | KNIFE (In+Out) | 60.47 (±0.81) | 62.39 (±0.42) |
| GPT-NeoX | CoT (I→RO) | 33.80 | 55.31 |
| GPT-3 (text-davinci-003) | CoT (I→RO) | 86.40 | 66.53 |

Table 5: **KNIFE low-resource learning results**

| Method | OBQA Acc. (↑) |
|---|---|
| FT (I→O) | 44.87 (±0.23) |
| FT (I→OR) | 38.07 (±1.14) |
| FT (I→RO) | 38.47 (±2.72) |
| FT (IR→O) | 44.80 (±2.23) |
| FT Dropout (IR→O) | 47.00 (±1.00) |
| FT Teacher Init. | 44.20 (±2.95) |
| KNIFE (In) | 47.20 (±1.40) |
| KNIFE (Out) | 48.13 (±2.12) |
| KNIFE (In+Out) | 47.47 (±1.96) |

## A.3 HYPERPARAMETERS

We always take AdamW as the optimizer. We stop training when the model performance on the development set has not improved for five epochs. The maximum epoch is 10.

For OBQA with T5-Base and Flan-T5-Base, we train the teacher model with learning rate of $1e{-}4$ and batch size of 64. We train the student model (by KD) with batch size of 64. For OBQA with T5-Large, we train the teacher model with learning rate of $5e{-}5$ and batch size of 64. We train the student model with batch size of 48. For the KD training student model, we always search the learning rate in $\{1e{-}4, 2e{-}4, 3e{-}4, 4e{-}4, 5e{-}4\}$.

For StrategyQA, we always set the warmup rate as 0.06. For T5-Base and Flan-T5-Base, we train the teacher model with learning rate of $3e{-}4$ and batch size of 16. For T5-Large, we train the teacher model with learning rate of $5e{-}5$ and batch size of 16. For the KD training student model, the batch size is always 16, and we always search the learning rate in $\{1e{-}4, 2e{-}4, 3e{-}4, 4e{-}4, 5e{-}4\}$.

FT (I→O), FT (I→OR), FT (I→RO), and FT (IR→O) use the same hyperparameters as the teacher models in the same settings do, except that we always search the learning rate in $\{1e{-}5, 2e{-}5, 5e{-}5\ 1e{-}4, 2e{-}4, 3e{-}4, 4e{-}4, 5e{-}4\}$. FT Dropout (IR→O) uses the same hyperparameters as FT (IR→O) does, and we search the dropout rate in $\{0.1, 0.2, 0.3, 0.4, 0.5, 0.6, 0.7, 0.8, 0.9\}$. FT Teacher Init. uses the same hyperparameters as FT (I→O) does.

Table 6: **Ablation study of FTR usage**

| Method | Accuracy (↑) | |
|---|---|---|
| | OBQA | StrategyQA |
| Non-FTR KD (Logit) | 48.53 (±4.06) | 57.72 (±2.12) |
| Non-FTR KD (In) | 31.87 (±2.10) | 53.77 (±0.46) |
| KNIFE (In) + Gold | 31.13 (±2.87) | 53.77 (±0.46) |
| KNIFE (In) + GPT-NeoX | 30.07 (±2.97) | 53.91 (±0.69) |
| Non-FTR KD (Out) | 55.60 (±2.99) | 59.99 (±3.53) |
| KNIFE (Out) + Gold | 55.60 (±2.42) | 61.12 (±0.53) |
| KNIFE (Out) + GPT-NeoX | 55.60 (±2.03) | 60.59 (±0.61) |
| Non-FTR KD (In+Out) | 58.27 (±1.01) | 60.12 (±1.40) |
| KNIFE (In+Out) + Gold | 60.93 (±0.12) | 61.12 (±2.03) |
| KNIFE (In+Out) + GPT-NeoX | 60.47 (±0.81) | 62.39 (±0.42) |

Table 7: **Ablation study of FTR quality**

| FTR Type | KNIFE Variant | Accuracy (↑) | |
|---|---|---|---|
| | | OBQA | StrategyQA |
| Replace | KNIFE (In) | 57.47 (±0.42) | 56.65 (±1.75) |
| Shuffle | KNIFE (In) | 57.73 (±1.01) | 56.65 (±2.71) |
| Gold | KNIFE (In) | 60.00 (±0.40) | 61.39 (±1.90) |
| GPT-NeoX | KNIFE (In) | 61.07 (±0.12) | 61.92 (±1.74) |
| Replace | KNIFE (Out) | 58.67 (±1.10) | 54.31 (±2.84) |
| Shuffle | KNIFE (Out) | 58.87 (±1.50) | 57.11 (±1.64) |
| Gold | KNIFE (Out) | 62.27 (±1.01) | 59.05 (±1.22) |
| GPT-NeoX | KNIFE (Out) | 61.60 (±0.53) | 60.72 (±0.20) |
| Replace | KNIFE (In+Out) | 58.87 (±1.30) | 55.44 (±4.43) |
| Shuffle | KNIFE (In+Out) | 59.07 (±0.31) | 56.91 (±1.59) |
| Gold | KNIFE (In+Out) | 61.53 (±0.76) | 60.45 (±0.31) |
| GPT-NeoX | KNIFE (In+Out) | 61.53 (±0.76) | 61.92 (±1.04) |
| Replace | KNIFE Teacher | 57.73 (±0.61) | 55.31 (±3.03) |
| Shuffle | KNIFE Teacher | 56.40 (±1.20) | 56.05 (±2.93) |
| Gold | KNIFE Teacher | 73.80 (±0.60) | 66.20 (±1.10) |
| GPT-NeoX | KNIFE Teacher | 74.33 (±0.46) | 64.93 (±1.40) |

## A.4 ABLATION STUDIES (EXTENDED)

We present the full results of ablation studies in the Appendix. Table 6 shows the full results of ablation studies on FTR usage. Table 7 shows the full results of ablation studies on FTR perturbation. Table 8 shows the full results of ablation studies on teacher bottleneck. Table 9 shows the full results of ablation studies on student task loss. The details of ablation studies are in 4.5. In the last three tables, we always use T5-Base→T5-Base.

Table 8: **Ablation study of teacher bottleneck**

| Bottleneck | KNIFE Variant | Accuracy (↑) | |
|---|---|---|---|
| | | OBQA | StrategyQA |
| No | KNIFE (In) | 58.67 (±0.70) | 49.77 (±2.91) |
| Yes | KNIFE (In) | 60.00 (±0.40) | 61.39 (±1.90) |
| No | KNIFE (Out) | 62.20 (±0.72) | 59.92 (±0.87) |
| Yes | KNIFE (Out) | 62.27 (±1.01) | 59.05 (±1.22) |
| No | KNIFE (In+Out) | 58.47 (±0.83) | 56.91 (±2.31) |
| Yes | KNIFE (In+Out) | 61.53 (±0.76) | 60.45 (±0.31) |
| No | KNIFE Teacher | 73.40 (±1.51) | 67.47 (±0.42) |
| Yes | KNIFE Teacher | 73.80 (±0.60) | 66.20 (±1.10) |

Table 9: **Ablation study of student task loss**

| Task Loss | KNIFE Variant | Accuracy (↑) | |
|---|---|---|---|
| | | OBQA | StrategyQA |
| Yes | KNIFE (In) | 59.73 (±1.10) | 56.31 (±1.00) |
| No | KNIFE (In) | 60.00 (±0.40) | 61.39 (±1.90) |
| Yes | KNIFE (Out) | 58.53 (±1.47) | 58.65 (±2.02) |
| No | KNIFE (Out) | 62.27 (±1.01) | 59.05 (±1.22) |
| Yes | KNIFE (In+Out) | 60.40 (±1.04) | 59.32 (±0.35) |
| No | KNIFE (In+Out) | 61.53 (±0.76) | 60.45 (±0.31) |

## A.5 IMPLEMENTATION DETAILS

For T5-Base, the parameter number of a single backbone model is around 220M. As we have two backbone models for the teacher and student model, the total number is around 440M. For T5-Large, the parameter number of a single backbone model is around 770M, and the total number is around 1.5B. GPT-NeoX has 20B parameters. We use NVIDIA Quadro RTX8000 for all experiments, which take around 700 GPU hours. We implement the models by HuggingFace Transformers. We also heavily use PyTorch and PyTorch Lightning.

## A.6 CASE STUDY OF FTR QUALITY

We give a representative example of a good OBQA FTR: "***Question**: There is most likely going to be fog around: **Answer Choices**: (A) a marsh, (B) a tundra, (C) the plains (D) a desert. **Gold FTR**: fog is formed by water vapor condensing in the air.*" The FTR explains the condition for fog formation. To answer this question with the FTR, one can identify as the answer a place where the condition (water vapor) is strong. The FTR describes the reasoning for answering the question.

We give a representative example of a bad ECQA FTR: "***Question**: What might a person see at the scene of a brutal killing? **Answer Choices**: (A) bloody mess, (B) pleasure, (C) being imprisoned, (D) feeling of guilt, (E) cake. **Gold FTR**: Bloody mess is covered or stained with blood. A person might see a bloody mess at the scene of a brutal killing.*" The first sentence of FTR just describes "bloody mess" from its literal meaning. The second sentence just fills in the answer to the question and rephrases the result into a declarative sentence. Overall, the FTR uninformatively states "bloody mess" is correct without explaining why.

We give a representative example of a bad QuaRTz FTR: "***Question**: If a moving object slows down, it will have ( ) kinetic energy. **Answer Choices**: (A) more, (B) less. **Gold FTR**: Anything that is moving has kinetic energy, and the faster it is moving, the more kinetic energy it has.*" The FTR states the outcome of an object speeding up without explanation or reasoning, and the answer is just the opposite of the outcome as the question asks the outcome of an object slowing down.

## A.7 FTR REPHRASING STUDY

Bad FTRs tend to simply rephrase the question and answer in an uninformative way (§4.6 and §A.6). Interestingly, for such FTRs, humans often can easily get the correct answer given only the FTR without the question. Based on this observation, we train a T5-Base to do FT (R→O), *i.e.,* to predict the answer taking only the FTR as the input on OBQA and ECQA. We also conduct this experiment on e-SNLI (Camburu et al., 2018), a dataset that has been proven to have very uninformative FTRs (Chen et al., 2022). We omitted StrategyQA and QuaRTz (where an instance has two opposite labels) in this analysis because their FTRs could be used to explain both the correct answer and the opposite label. The result is shown in Table 10.[8] We find that ECQA and e-SNLI tend to do simple rephrasing much more than OBQA does, which further proves the bad quality of ECQA and e-SNLI.[9]

---

[8] The accuracy of Random Guessing is the inverse of the number of labels.

[9] Note that this experiment is not a comprehensive quantitative analysis as dataset analysis is not our focus.

Table 10: **FTR rephrasing study**

| Method | Accuracy (↑) | | |
|---|---|---|---|
| | OBQA | ECQA | e-SNLI |
| Random Guessing | 25.00 | 20.00 | 33.33 |
| FT (R→O) | 58.53 (±1.03) | 97.69 (±0.32) | 96.03 (±0.42) |

Table 11: **KNIFE results for FLAN-T5-Base**

| Architecture | Method | Accuracy (↑) | |
|---|---|---|---|
| | | OBQA | StrategyQA |
| T5-Base→T5-Base | FT (I→O) | 63.80 (±0.53) | 63.66 (±0.70) |
| | FT (I→OR) | 62.87 (±1.01) | 57.25 (±1.86) |
| | FT (I→RO) | 62.27 (±0.12) | 62.79 (±3.41) |
| | FT (IR→O) | 59.07 (±0.42) | 55.58 (±3.20) |
| | FT Dropout (IR→O) | 63.27 (±0.64) | 60.92 (±2.03) |
| | FT (I→R, IR'→O) | 61.93 (±1.14) | 53.64 (±4.37) |
| | FT Teacher Init. | 64.07 (±0.46) | 61.00 (±1.91) |
| | KNIFE (In) | 64.80 (±0.40) | 62.59 (±1.56) |
| | KNIFE (Out) | 65.53 (±2.04) | 62.26 (±2.23) |
| | KNIFE (In+Out) | 64.27 (±1.51) | 62.26 (±1.42) |

## A.8 ETHICS STATEMENT

Regarding ethical concerns, all datasets used in this work are publicly available. Besides using gold rationales, our method can make use of large language models to generate free-text rationales. We are aware that the resulting rationales may contain social biases and that such biases may be further inherited by the student model.

Table 12: **KNIFE teacher results (T5-Base)**

| Model | Using FTRs during inference | Accuracy (↑) | |
|---|---|---|---|
| | | OBQA | StrategyQA |
| FT (I→O) | ✗ | 57.93 (±1.15) | 59.05 (±0.23) |
| FT (IR→O) | ✗ | 53.73 (±2.31) | 49.97 (±2.92) |
| KNIFE Teacher | ✗ | 51.27 (±1.86) | 50.37 (±2.03) |
| | ✓ | 73.80 (±0.60) | 66.20 (±1.10) |

Table 13: **Ablation study of FTR usage (T5-Base)**

| Method | Accuracy (↑) | |
|---|---|---|
| | OBQA | StrategyQA |
| FT (I→O) | 57.93 (±1.15) | 59.05 (±0.23) |
| Non-FTR KD (Logit) | 58.87 (±2.14) | 59.25 (±1.70) |
| Non-FTR KD (In + Out) | 59.47 (±0.12) | 58.98 (±0.83) |
| KNIFE (In+Out) + Gold | 60.93 (±0.12) | 61.12 (±2.03) |
| KNIFE (In+Out) + GPT-NeoX | 60.47 (±0.81) | 62.39 (±0.42) |

