# OpenReview forum: "KNIFE: Distilling Reasoning Knowledge From Free-Text Rationales"
_ICLR.cc/2024/Conference — Submitted to ICLR 2024_

### Official Review · Reviewer_HVcW · 2023-10-29

**Soundness:** 2 fair
**Presentation:** 3 good
**Contribution:** 2 fair
**Rating:** 3
**Confidence:** 3

**Summary:**

This paper explores the idea of distilling knowledge from free-text explanations. They start by training a "teacher" model, which learns to make predictions using both the free-text explanations and task inputs. This helps the teacher model absorb the knowledge from the free-text explanations into its hidden state. Next, they build a "student" model and make sure its hidden state aligns with the teacher's, all without directly using the free-text explanations. They then put their models to the test on two question-answering tasks, comparing them to various baseline models with different fine-tuning and initialization approaches. Impressively, their models outperform these baselines. Additionally, they carry out an ablation study to understand how different parts of their model contribute to its success.

**Strengths:**

They study the idea of knowledge distillation from free-text rationales and did comprehensive experiments to show the effectiveness of the approach.

**Weaknesses:**

* The idea behind this approach isn't very convincing. The teacher model can't store a lot of knowledge, and it might not work well for different tasks. Plus, it's unclear how this method is better than retrieval-augmented generation.
* To prove its effectiveness, more experiments should be done, comparing it to retrieval-augmented generation and testing it on various downstream tasks.
* The improvement from introducing free-text rationale into the teacher model isn't substantial, and it might be because of the extra knowledge.
* Their best model still falls far short of the Large Language Models (LLM) by a significant margin, and the student model doesn't seem to improve model explanation and reasoning.

**Questions:**

1. What is the upper-bound performance achievable by the small model when given the task input and free-text rationale together?
2. How does this approach compare favorably to "retrieve-augmented-generation," a method that initially retrieves relevant knowledge, such as free-text rationales, and then enhances the input with this retrieved knowledge?
3. Why does incorporating the free-text rationale during the teacher model's fine-tuning enhance the student model's performance? It's possible that the improvement shown in Figure 3(a) is due to the extra knowledge contained in the free-text rationale, which could be beneficial for the downstream tasks. This improvement, while not substantial, might not generalize well to scenarios with less overlap in knowledge or when using larger free-text rationales for teacher model fine-tuning. This raises questions about the broader applicability of this approach.

---

> ### Author Response · Authors · 2023-11-16
> **Response to Reviewer HVcW**
>
> Thank you for your insightful and constructive feedback! We were encouraged that you see our comprehensive experiments and analysis. Below, we provide clarifications and address the concerns you raised.
>
> **(1) Retrieval-Augmented Approach as Baseline** (Weakness 1, Weakness 2, Question 2)
>
> Thanks for your helpful suggestions! Please refer to the general response for more details.
>
> **(2) Justification for KNIFE Design** (Weakness 1, Question 3)
>
> The teacher model is indeed capable of storing task-level reasoning knowledge. This is empirically evidenced in Table 12 (see more discussions below), where the teacher model demonstrates very high performance.
>
> Regarding the improvement of the student model's performance through the incorporation of FTRs during the teacher's training: Our intuition is that the teacher's finetuning aggregates reasoning knowledge from FTRs across all instances, subsequently embedding this knowledge in the teacher’s hidden states. KNIFE leverages these enriched hidden states as soft labels, enabling the distillation of reasoning knowledge from the teacher to the student via the collection of all soft labels. During the student model’s inference, its output is not explicitly conditioned on specific FTRs, but rather, it is implicitly influenced by the distilled reasoning knowledge embedded within the student's parameters.
>
> **(3) Upper-Bound Performance** (Question 1)
>
> The upper-bound performance is achieved by providing the task inputs and FTRs to models during both training and inference, which is reflected in the teacher model’s performance when using FTRs during inference as reported in Table 12. Compared to FT (I→O), the accuracy improvement is approximately 16% and 7% on OpenbookQA and StrategyQA, respectively, which is quite substantial. It's important to note that this performance should not be compared to results where the FTRs are not accessible during inference.
>
> **(4) Teacher Model Performance** (Weakness 3)
>
> While you suggest that the improvement of the teacher model might be due to extra knowledge, we posit that this is an inherent advantage of our approach. As discussed, a set of task instances that sufficiently characterizes a task implies that a set of FTRs for these instances can collectively capture task-level reasoning knowledge that generalizes to unseen task instances. This task-level reasoning knowledge is transferred to the teacher by finetuning it on both the task inputs and the FTRs. Previous works on small models failed to extract such extra knowledge. By contrast, KNIFE successfully improves the model performance by effectively extracting extra knowledge provided by FTRs.
>
> **(5) Performance Improvement** (Weakness 1, Weakness 4)
>
> We conducted the Wilcoxon rank-sum test between KNIFE and the best baseline for each OBQA/StrategyQA setting, and for both human-written and model-annotated FTR sources. KNIFE achieves statistically significant improvements (p<0.05) over the best baseline in all settings, except in the T5-Large→T5-Large on StrategyQA.
>
> We also believe that the comparison to large-scale language models (LLMs) is unfair, as they are often approximately 100 times larger than the LMs we experimented with. Our focus is on how to improve the performance of smaller LMs. To our knowledge, previous works have even failed to improve small-scale LMs by incorporating FTRs. We believe KNIFE represents an important contribution to the field of learning from FTRs.
>
> **(6) FTRs’ Influence on KNIFE’s Performance** (Question 3)
>
> We appreciate your insights regarding the influence of FTRs on KNIFE's performance. Indeed, since KNIFE distills knowledge from FTRs into the student language model, the student's performance is expected to depend on the quantity and quality of reasoning knowledge contained in the FTRs. You are correct in noting that if the FTRs are uninformative (in terms of providing task-level reasoning knowledge), KNIFE's ability to enhance model performance is limited. We have provided additional analysis and case studies in Section 4.6.

---

### Official Review · Reviewer_MPYM · 2023-11-01

**Soundness:** 3 good
**Presentation:** 3 good
**Contribution:** 2 fair
**Rating:** 5
**Confidence:** 4

**Summary:**

This paper proposes KNowledge dIstillation from Free-text rationalEs (KNIFE), a distillation method that leverages open-source pretrained seq2seq models and training data with free-text rationale (FTR) annotations to develop an accurate question-answering prediction model. It first finetunes a teacher LM to input question and FTR and output the answer. Then it finetunes a student to input the question and output the answer, while aligning to the teacher's hidden states. The student outperforms finetuning and prompting baselines in fully-supervised and low-resource settings. The paper further shows that the FTR quality is important to the success.

**Strengths:**

- The scenario is likely: there are rationale annotations for training data, but these annotations are not readily available at test time.
- The method is intuitive and effective. The bottleneck architecture has novelty.
- Compared with multiple baselines and performed careful ablation study.

**Weaknesses:**

- The paper is about classification and shows advantages on two datasets. Results on more tasks will be helpful for showing the generality of the method. Does the method work for reasoning tasks commonly used by the chain-of-thought literature, such as, arithmetic reasoning, commonsense reasoning, and code generation? Does it work for knowledge-intensive tasks?
- The paper doesn't have a retrieval-augmentation baseline. Will the numbers look better if you finetune T5 to learn to condition on retrieved (question, rationale, answer) demonstrations, instead of distilling rationales from text to a teacher and then to a student?

**Questions:**

Please refer to the Weaknesses section.

---

> ### Author Response · Authors · 2023-11-16
> **Response to Reviewer MPYM**
>
> Thank you for your insightful and constructive feedback! We were encouraged that you see KNIFE’s novelty, effectiveness, and usefulness and our comprehensive experiments and analysis. Below, we provide clarifications and address the concerns you raised.
>
> For the task type, KNIFE’s design is generic enough that it could be extended to generation tasks. Classification and generation tasks differ primarily in their output formats, but KNIFE’s key components (i.e., teacher bottleneck and teacher-student hidden states distillation) do not involve the LM’s final output specifically. We leave the extension of KNIFE to generation tasks for future work.
>
> For the retrieved-augmented baselines, thanks for your helpful suggestions! Please refer to the general response for more details.

---

> > ### Comment · Reviewer_MPYM · 2023-12-03
> >
> > Thank you for the response and additional experiments. I share concerns with other reviewers about the completeness of the experiments. In particular, it would be great if the next version could include commonly used reasoning tasks mentioned in the original review.

---

### Official Review · Reviewer_izvw · 2023-11-01

**Soundness:** 2 fair
**Presentation:** 3 good
**Contribution:** 1 poor
**Rating:** 3
**Confidence:** 4

**Summary:**

This work introduces KNowledge DIstillation From Free-Text RationalEs (KNIFE) as a method to effectively distill the reasoning knowledge from a large Language Model (LM) to a smaller LM, aiming to enhance the smaller LM’s task performance. Specifically, the teacher model is fine-tuned to predict the answer based on the task input (question) and the pre-defined free-text raionales associated with each training data. Then, the hidden states of the encoder and output prediction distribution are used in knowledge distillation, transferring knowledge from the teacher model to the student model. Experimental results indicate that KNIFE is effective compared to the fine-tuning variants thanks to the knowledge distillation of both hidden states and output distribution.

**Strengths:**

- **New Approach to a Specific Problem;** The proposed method represents a novel approach to a specialized problem where either human-written or machine-generated free-text rationales are available, and the language model architecture is based on an encoder-decoder system like T5. As far as I am aware,  this particular issue hasn’t been addressed in previous works.
- **Thorough Analysis;** The authors conducted comprehensive experiments under various conditions, including two different sizes of LM architectures, two distinct datasets, varying input-output compositions, and FTR variants.

**Weaknesses:**

- **Limited Contribution;** While I acknowledge the novelty and design of the prospoed method aimed at distilling reasoning knowledge from free-text rationales in encoder-decoder LMs, its contribution appears limited for several reasons:
    - The efficacy of the proposed method, KNIFE, seems marginal as the improvements are not statistically significant based on some results in Table 1. For instance, in the StrategyQA dataset, KNIFE’s performance is comparable to FT (I→RO), suggesting that simple fine-tuning of the language model with FTR and using the answer as the target yields results similar to KNIFE. Additionally, the experimental results suggest that distilling from T5-Large to T5-Base is less effective than from T5-Base to T5-base, which is weird.
    - The application scope of the proposed method is restricted both in task type and language model architecture. It’s only suitable for multi-choice QA tasks when free-text rationales are available and is exclusive to LMs with encoder-decoder architecture like T5, making it unsuitable for decoder-only models like Llama.
    - The individual contributions of each component in the proposed method remain ambiguous. In Tables 8 and 9, KNIFE with either KD-in loss only or KD-out loss only occasionally outperforms KNIFE with both objectives combined. This raises the question: is employing both objectives truly advantageous?
- **Limited significance of the Problem;** Lately, there have been significant works [1,2,3] into distilling the reasoning ability of large LMs (e.g., Llama-2 or GPT-3.5-turbo) into smaller LMs. Given this context, how does the proposed method compare to existing approaches? In Appendix A.1., the authors posit that their method is advantageous when large-scale LMs lack reasoning abilities, a premise that seems out of sync with current trends in large language models.

[1] Ho et al., Large language models are reasoning teachers, ACL 2023

[2] Magister et al., Teaching small language models to reason, ACL 2023

[3] Fu et al., Specializing smaller language models towards multi-step reasoning, ICML 2023

**Questions:**

1. What is the advantage of this work compared to recent CoT distillation works mentioned in the second point of weaknesses?
2. Why do the authors not include results with In only and out only in Table 1? I think this baseline is important to show the significance of using both objectives in KNIFE.

---

> ### Author Response · Authors · 2023-11-16
> **Response to Reviewer izvw [1/2]**
>
> Thank you for your insightful and constructive feedback! We were encouraged that you see KNIFE’s great novelty and our comprehensive experiments and analysis. Below, we provide clarifications and address the concerns you raised.
>
> **(1) Performance Improvement**
>
> We conducted the Wilcoxon rank-sum test between KNIFE and the best baseline for each OBQA/StrategyQA setting, and for both human-written and model-annotated FTR sources. KNIFE achieves statistically significant improvements (p<0.05) over the best baseline in all settings, except in the T5-Large→T5-Large on StrategyQA. By contrast, previous works, to our knowledge, have even failed to improve small-scale LMs by incorporating FTRs. We believe KNIFE represents an important contribution to the field of learning from FTRs.
>
> **(2) Comparison between T5-Base→T5-Base and T5-Large→T5-Base**
>
> We agree with your observation. It is indeed true that distilling from T5-Large to T5-Base (T5-Large→T5-Base) is less effective than distilling from T5-Base to T5-Base (T5-Base→T5-Base) in some settings, and this is not unusual. Our paper states an important implementation detail in Section 4.2: when the student and teacher models both use the same backbone (i.e., either T5-Base or T5-Large), the student’s parameters are initialized with the teacher’s. Additionally, T5-Base and T5-Large have different representation spaces in their hidden states, and since we perform distillation via these hidden states (as mentioned in footnote 4), there is an added complexity. We stated that we jointly train two linear projection layers to transform the hidden states for T5-Large→T5-Base distillation, while T5-Base→T5-Base distillation does not require this. These factors contribute to the increased difficulty in transferring reasoning knowledge from T5-Large to T5-Base compared to T5-Base to T5-Base. Although a T5-Large teacher model is expected to be stronger, the complexity of the distillation process also impacts performance. Due to the page limit, we will include a more comprehensive discussion on this topic in our next version.
>
> **(3) Application Scope**
>
> KNIFE’s design is generic enough that it could be extended to generation tasks. Classification and generation tasks differ primarily in their output formats, but KNIFE’s key components (i.e., teacher bottleneck and teacher-student hidden states distillation) do not involve the LM’s final output specifically. We leave the extension of KNIFE to generation tasks for future work.
>
> Additionally, the availability of FTRs during the training stage is a fundamental assumption in our study. We have discussed the feasibility of this setting in the Introduction section of our paper.
>
> **(4) Model Architecture**
>
> Our paper follows prior FTR-based works [1-3] in focusing on encoder-decoder LMs like T5, which has been shown to achieve strong “out-of-the-box” finetuning performance across a wide range of different text classification and generation tasks as well as produce its own FTRs.
>
> Although the teacher bottleneck is currently tailored for encoder-decoder LMs (since it operates in the cross-attention mechanism), the bottleneck could potentially be modified to work for decoder-only or encoder-only LMs (e.g., GPT, LLaMA, BERT) too. For these LMs, one possible bottleneck design would be to mask out the self-attention weights for all FTR tokens in the teacher LM’s last k (e.g., k=1) layers, then perform KD only via the hidden states of the task input tokens in these last k layers. However, due to computational constraints, we leave investigation of decoder-only/encoder-only KNIFE designs to future work.
>
> **(5) Individual Contributions of KNIFE (In) and KNIFE (Out)**
>
> To clarify, both KNIFE (In) and KNIFE (Out) are novel contributions of this work and not main baselines used for comparison. We refer readers to Table 2 (instead of Tables 8 or 9) for a comparison among KNIFE (In), KNIFE (Out), and KNIFE (In+Out). We observe that KNIFE (In+Out) generally achieves the highest performance, with a few exceptions. This indicates that valuable FTR knowledge can be distilled through both encoder and decoder hidden states, thus we recommend using both by default. Additionally, KNIFE (In) and KNIFE (Out) outperform all baselines in almost all cases. We included only KNIFE (In+Out) in the main table for clarity of presentation, aiming to convey the message that “KNIFE can outperform all baselines” without distracting readers. Then, a detailed comparison among the three KNIFE variants is provided in the following part of our paper. We will improve our presentation to better convey our messages in our next version.
>
> Table 8 mainly presents results for KNIFE without the bottleneck design, which is not used for comparing the three KNIFE variants as the bottleneck is a critical design feature of KNIFE. Table 9 focuses on results for KNIFE with task loss, which is also not used in the comparison since incorporating task loss degrades KNIFE’s performance.

---

> ### Author Response · Authors · 2023-11-16
> **Response to Reviewer izvw [2/2]**
>
> **(6) Relevant Works about CoT Distillation**
>
> To clarify, our method, KNIFE, is not predicated on the absence of reasoning abilities in large-scale LMs. Instead, one advantage actually lies in its independence from LLMs. This allows KNIFE to work effectively without relying on the computational resources required for large LMs. Regarding the comparison with the approaches mentioned in your review and our Appendix A.1, it is important to note that these methods leverage large LMs to enhance smaller models, whereas KNIFE utilizes FTRs. This difference in approach makes fair comparison challenging.
>
> In fact, we posit that relying on large LMs, as these methods do, is a stronger assumption compared to access to FTRs, as people can employ LLMs for generating FTRs. We have experimented with generating FTRs using a medium-sized LM, GPT-neox, and found KNIFE effective in this context. Due to computational constraints, exploring this further with larger LMs like ChatGPT and LLaMA-70B remains a goal for future research.
>
> Most of the referenced works can be seen as utilizing LMs for FTR generation and training another model in I→RO. Our findings indicate that the I→RO is not effective with the FTRs we have experimented with. This observation leads us to believe that the **generation of high-quality FTRs** by LLMs is crucial in those approaches. As our work focuses on the **utilization of FTRs**, positioning it also as complementary to those approaches.
>
> We will add more discussion on the points above in our next version.
>
> [1] Narang et al. “WT5?! Training Text-to-Text Models to Explain their Predictions.” 2020.
>
> [2] Sun et al. “Investigating the Benefits of Free-Form Rationales.” 2022.
>
> [3] Wiegreffe et al. “Measuring Association Between Labels and Free-Text Rationales.” 2021.

---

> ### Comment · Reviewer_izvw · 2023-11-23
>
> Thank you for your detailed response. After reading the author's reply and other reviews, my confidence has grown that this paper is not yet ready for publication. Specifically, while I acknowledge the statistical significance of the performance improvement, I question the applicability of the proposed method in the various scenarios mentioned by the authors, in the absence of adequate empirical evidence. Beyond the novelty and performance enhancement of the proposed method, I believe the experiments are still too limited, leaving readers unconvinced of the method's merits. I hope the authors will address this gap in their next revision.

---

### Author Response · Authors · 2023-11-16
**General Response**

We sincerely thank all the reviewers for their thoughtful feedback and recognition of our paper’s contributions. We are delighted to see that the reviewers appreciated the novelty, experiments, and motivation of KNIFE.

A common concern, noted by both reviewers MPYM and HVcW, is the lack of comparison with a retrieval-augmented baseline. We greatly appreciate this constructive feedback and have conducted the suggested experiments. Specifically, we trained a model to solve the task, conditioned on retrieved extra information. As the extra information used by KNIFE consists of the free-text rationales of training instances, we used this collection as the retrieval corpus for fair comparison. For the retriever, we chose BM25, due to its proven efficiency and effectiveness in information retrieval tasks. The results are as follows:

|  Model   |          Method           | OpenbookQA (Accuracy) | StrategyQA (Accuracy) |
| :------: | :-----------------------: | :-------------------: | :-------------------: |
| T5-Base  | FT (IR→O, R is retrieved) |    59.13  (±0.31)     |     54.38 (±2.42)     |
| T5-Base  |      KNIFE (In+Out)       |     61.53 (±0.76)     |     60.45 (±0.31)     |
| T5-Large | FT (IR→O, R is retrieved) |     68.53 (±0.76)     |    57.78  (±2.23)     |
| T5-Large |      KNIFE (In+Out)       |     68.73 (±1.36)     |     63.79 (±0.64)     |

From our results, it is evident that KNIFE generally outperforms the retrieval-augmented baseline by a significant margin, particularly on StrategyQA. This demonstrates that KNIFE can effectively extract task-level reasoning knowledge from all FTRs, thereby enhancing task performance. In contrast, the retrieval-augmented method fails to effectively utilize the collective FTRs of all training instances. Due to the page limit, we will include this baseline in the next version of our paper.

---

### Meta-Review · Area_Chair_th9G · 2023-12-20

**Metareview:**

This work propose a method (KNIFE) to distill reasoning knowledge from large language model to a smaller LM. The reasoning knowledge is extracted using pre-defined rationale templates.

The main concerns are: The novelty of the paper is limited since it does not go beyond knowledge distillation. The results on Table 1 do not show consistent significant improvement. It is also limited to the type of problems.

**Justification For Why Not Higher Score:**

The novelty of the paper is limited since it does not go beyond knowledge distillation. The results on Table 1 do not show consistent significant improvement. It is also limited to the type of problems.

**Justification For Why Not Lower Score:**

N/A

---

### Decision · Program_Chairs · 2024-01-16

Reject